# Graph Convolution with Low-rank Learnable Local Filters

**Xiuyuan Cheng**
Department of Mathematics
Duke University
Durham, NC, USA
`xiuyuan.cheng@duke.edu`

**Zichen Miao & Qiang Qiu**
Electrical and Computer Engineering
Purdue University
West Lafayette, IN, USA
`{miaoz,qqiu}@purdue.edu`

## Abstract

Geometric variations like rotation, scaling, and viewpoint changes pose a significant challenge to visual understanding. One common solution is to directly model certain intrinsic structures, e.g., using landmarks. However, it then becomes non-trivial to build effective deep models, especially when the underlying non-Euclidean grid is irregular and coarse. Recent deep models using graph convolutions provide an appropriate framework to handle such non-Euclidean data, but many of them, particularly those based on global graph Laplacians, lack expressiveness to capture local features required for representation of signals lying on the non-Euclidean grid. The current paper introduces a new type of graph convolution with learnable low-rank local filters, which is provably more expressive than previous spectral graph convolution methods. The model also provides a unified framework for both spectral and spatial graph convolutions. To improve model robustness, regularization by local graph Laplacians is introduced. The representation stability against input graph data perturbation is theoretically proved, making use of the graph filter locality and the local graph regularization. Experiments on spherical mesh data, real-world facial expression recognition/skeleton-based action recognition data, and data with simulated graph noise show the empirical advantage of the proposed model.

## 1 Introduction

Deep methods have achieved great success in visual cognition, yet they still lack capability to tackle severe geometric transformations such as rotation, scaling and viewpoint changes. This problem is often handled by conducting data augmentations with these geometric variations included, e.g. by randomly rotating images, so as to make the trained model robust to these variations. However, this would remarkably increase the cost of training time and model parameters. Another way is to make use of certain underlying structures of objects, e.g. facial landmarks (Chen et al., 2013) and human skeleton landmarks (Vemulapalli et al., 2014a), c.f. Fig. 1 (right). Nevertheless, these methods then adopt hand-crafted features based on landmarks, which greatly constrains their ability to obtain rich features for downstream tasks. One of the main obstacles for feature extraction is the non-Euclidean property of underlying structures, and particularly, it prohibits the direct usage of prevalent convolutional neural network (CNN) architectures (He et al., 2016; Huang et al., 2017). Whereas there are recent CNN models designed for non-Euclidean grids, e.g., for spherical mesh (Jiang et al., 2019; Cohen et al., 2018; Coors et al., 2018) and manifold mesh in computer graphics (Bronstein et al., 2017; Fey et al., 2018), they mainly rely on partial differential operators which only can be calculated precisely on fine and regular mesh, and may not be applicable to the landmarks which are irregular and course. Recent works have also applied Graph Neural Network (GNN) approaches to coarse non-Euclidean data, yet methods using GCN (Kipf & Welling, 2016) may fall short of model capacity, and other methods adopting GAT (Veličković et al., 2017) are mostly heuristic and lacking theoretical analysis. A detailed review is provided in Sec. 1.1.

In this paper, we propose a graph convolution model, called *L3Net*, originating from low-rank graph filter decomposition, c.f. Fig. 1 (left). The model provides a unified framework

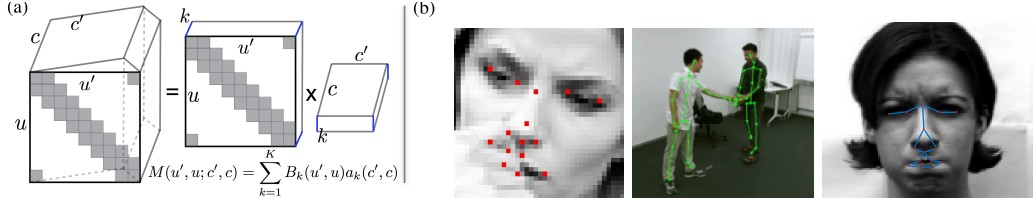

Figure 1: (a) $K$-rank graph local filters. Notation as in Sec. 2.1, and specifically, $u$ is node index, $c$ is channel index, $k$ is basis index, and $K$ is number of basis. $M$ is the tensor in the GNN linear mapping (1) (2), decomposed into learnable local basis $B_k$ combined by learnable coefficients $a_k$. (b) The first two figures shows the good property of landmarks for being invariant to pose and camera viewpoint changes. The third figure illustrates the graph we built on facial landmarks.

for graph convolutions, including ChebNet (Defferrard et al., 2016), GAT, EdgeNet (Isufi et al., 2020) and CNN/geometrical CNN with low-rank filter as special cases. In addition, we theoretically prove that L3Net is strictly more expressive to represent graph signals than spectral graph convolutions based on global adjacency/graph Laplacian matrices, which is then empirically validated, c.f. Sec. 3.1. We also prove a Lipschitz-type representation stability of the new graph convolution layer using perturbation analysis.

Because our model allows neighborhood specialized local graph filters, regularization may be needed to prevent over-fitting, so as to handle changing underlying graph topology and other graph noise, e.g., inaccurately detected landmarks or missing landmark points due to occlusions. Therefore, we also introduce a regularization scheme based on local graph Laplacians, motivated by the eigen property of the latter. This further improves the representation stability aforementioned. The improved performance of L3Net compared to other GNN benchmarks is demonstrated in a series of experiments, and with the the proposed graph regularization, our model shows robustness to a variety of graph data noise.

In summary, the contributions of the work are the following:

- We propose a new graph convolution model by a low-rank decomposition of graph filters over trainable local basis, which unifies several previous models of both spectral and spatial graph convolutions.
- Regularization by local graph Laplacians is introduced to improve the robustness against graph noise.
- We provide theoretical proof of the enlarged expressiveness for representing graph signals and the Lipschitz-type input-perturbation stability of the new graph convolution model.
- We demonstrate with applications to object recognition of spherical data and facial expression/skeleton-based action recognition using landmarks. Model robustness against graph data noise is validated on both real-world and simulated datasets.

## 1.1 Related Works

**Modeling on face/body landmark data.** Many applications in computer vision, such as facial expression recognition (FER) and skeleton-based action recognition, need to extract high-level features from landmarked data which are sampled at irregular grid points on human face or at body joints. While CNN methods (Guo et al., 2016; Ding et al., 2017; Meng et al., 2017) prevail in FER task, landmark methods have the potential advantage in lighter model size as well as more robustness to previously mentioned geometric transformations like pose variation. Earlier methods based on facial landmarks used hand-crafted features (Jeong & Ko, 2018; Morales-Vargas et al., 2019) rather than deep networks. Skeleton-based methods in action recognition have been developed intensively recently (Ren et al., 2020), including non-deep methods (Vemulapalli et al., 2014b; Wang et al., 2012) and deep methods (Ke et al., 2017; Kim & Reiter, 2017; Liu et al., 2016; Yan et al., 2018). Facial and skeleton landmarks only give a coarse and irregular grid, and then mesh-based geometrical CNN's are hardly applicable, while previous GNN models on such tasks may lack sufficient expressive power.

**Graph convolutional network.** A systematic review can be found in several places, e.g. Wu et al. (2020). Spectral graph convolution was proposed using full eigen decomposition of the graph Laplacian in Bruna et al. (2013), Chebyshev polynomial in ChebNet (Defferrard

et al., 2016), by Cayley polynomials in Levie et al. (2018). GCN (Kipf & Welling, 2016), the mostly-used GNN, is a variant of ChebNet using degree-1 polynomial. Liao et al. (2019) accelerated the spectral computation by Lanczos algorithm. Graph scattering transform has been developed using graph wavelets (Zou & Lerman, 2020; Gama et al., 2019b), which can be constructed in the spectral domain (Hammond et al., 2011) and by diffusion wavelets (Coifman & Maggioni, 2006). The scattering transform enjoys theoretical properties of the representation but lacks adaptivity compared to trainable neural networks. Spatial graph convolution has been performed by summing up neighbor nodes' transformed features in NN4G (Scarselli et al., 2008), by graph diffusion process in DCNN (Atwood & Towsley, 2016), where the graph propagation across nodes is by the adjacency matrix. Graph convolution with trainable filter has also been proposed in several settings: MPNN (Gilmer et al., 2017) enhanced model expressiveness by message passing and sub-network; GraphSage (Hamilton et al., 2017) used trainable differential local aggregator functions in the form of LSTM or mean/max-pooling; GAT (Veličković et al., 2017) and variants (Li et al., 2018; Zhang et al., 2018; Liu et al., 2019) introduced attention mechanism to achieve adaptive graph affinity, which remains non-negative valued; EdgeNet (Isufi et al., 2020) developed adaptive filters by taking products of trainable local filters. Our model learns local filters which can take negative values and contains GAT and EdgeNet as special cases. Theoretically, expressive power of GNN has been studied in Morris et al. (2019); Xu et al. (2019); Maron et al. (2019a;b); Keriven & Peyré (2019), mainly focusing on distinguishing graph topologies, while our primary concern is to distinguish signals lying on a graph.

**CNN and geometrical CNN.** Standard CNN applies local filters translated and shared across locations on an Euclidean domain. To extend CNN to non-Euclidean domains, convolution on a regular spherical mesh using geometrical information has been studied in S2CNN (Cohen et al., 2018), SphereNet (Coors et al., 2018), SphericalCNN (Esteves et al., 2018), and UGSCNN (Jiang et al., 2019), and applied to 3D object recognition, for which other deep methods include 3D convolutional (Qi et al., 2016) and non-convolutional architectures (Qi et al., 2017a;b). CNN's on manifolds construct weight-sharing across local atlas making use of a mesh, e.g., by patch operator in Masci et al. (2015), anisotropic convolution in ACNN (Boscaini et al., 2016), mixture model parametrization in MoNet (Monti et al., 2017), spline functions in SplineCNN (Fey et al., 2018), and manifold parallel transport in Schonsheck et al. (2018). These geometric CNN models use information of non-Euclidean meshes which usually need sufficiently fine resolution.

## 2 METHOD

### 2.1 DECOMPOSED LOCAL FILTERS

Consider an undirected graph $G = (V, E)$, $|V| = n$. A graph convolution layer maps from input node features $X(u', c')$ to output $Y(u, c)$, where $u, u' \in V$, $c' \in [C']$ ($c \in [C]$) is the input (output) channel index, the notation $[m]$ means $\{1, \cdots, m\}$, and

$$Y(u, c) = \sigma(\sum_{u' \in V, c' \in [C']} M(u', u; c', c)X(u', c') + \text{bias}(c)), \quad u \in V, c \in [C]. \tag{1}$$

The spatial and spectral graph convolutions correspond to different ways of specifying $M$, c.f. Sec. 2.3. The proposed graph convolution is defined as

$$M(u', u; c', c) = \sum_{k=1}^{K} a_k(c', c)B_k(u', u), \quad a_k(c', c) \in \mathbb{R}, \tag{2}$$

where $B_k(u', u)$ is non-zero only when $u' \in N_u^{(d_k)}$, $N_u^{(d)}$ denoting the $d$-th order neighborhood of $u$ (i.e., the set of $d$-neighbors of $u$), and $K$ is a fixed number. In other words, $B_k$'s are $K$ basis of local filters around each $u$, and the order $d_k$ can differ with $1 \le k \le K$. Both $a_k$ and $B_k$ are trainable, so the number of parameters are $K \cdot CC' + \sum_{k=1}^{K} \sum_{u \in V} |N_u^{(d_k)}| \sim K \cdot CC' + Knp$, where $p$ stands for the average local patch size. In our experiments we use $K$ up to 5, and $d_k$ up to 3. We provide the matrix notation of (2) in Appendix A.1.

The construction (2) can be used as a layer type in larger GNN architectures. Pooling of graphs can be added between layers, and see Appendix C.5 for further discussion on multi-scale model. The choice of $K$ and neighborhood orders $(d_1, \cdots, d_K)$ can also be adjusted accordingly. The model may be extended in several ways to be discussed in the last section.

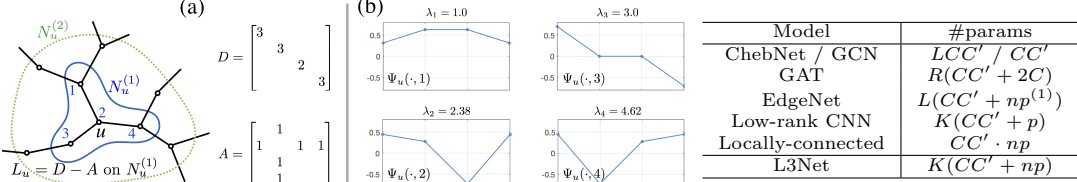

Figure 2: Plots: (a) Local graph Laplacian $L_u := D - A$ on a neighborhood around node $u$. (b) Plots of the Dirichlet eigenvectors on the local graph. The first Dirichlet eigenvector does not change sign on $N_u$ and is envelope-like. (Table) Model complexity measured by number of parameters, $C$ and $C'$ being the number of input and output channels, $p$ $(p^{(1)})$ the average patch size of local neighborhoods (local 1-neighborhoods), see more in Sec. 2.3.

## 2.2 Regularization by local graph Laplacian

The proposed L3Net layer enlarges the model capacity by allowing $K$ basis filters at each location, and a natural way to regularize the trainable filters is by the graph geometry, where, by construction, only the local graph patch is concerned. We introduce the following regularization penalty of the basis filters $B_k$'s as

$$\mathcal{R}(\{B_k\}_k) = \sum_{k=1}^{K} \sum_{u \in V} (b_u^{(k)})^T L_u^{(k)} b_u^{(k)}, \quad b_u^{(k)}(v) := B_k(v, u), \; b_u^{(k)} : N_u^{(d_k)} \to \mathbb{R}, \qquad (3)$$

where $L_u^{(k)}$, equaling $(D - A)$ restricted to the subgraph on $N_u^{(d_k)}$, is the Dirichlet local graph Laplacian on $N_u^{(d_k)}$ (Chung & Graham, 1997) (Fig. 2). The training objective is

$$\mathcal{L}(\{a_k, B_k\}_k) + \lambda \mathcal{R}(\{B_k\}_k), \quad \lambda \geq 0, \qquad (4)$$

where $\mathcal{L}$ is the classification loss. As $\mathcal{L}$ encourages the diversity of $B_k$'s, the $K$-rankness usually remains a tight constraint in training, unless $\lambda$ is very large, see also Proposition 3.

## 2.3 A unified framework for graph convolutions

Graph convolutions basically fall into two categories, the spatial and spectral constructions (Wu et al., 2020). The proposed L3Net belongs to spatial construction, and here we show that the model (2) is a unified framework for various graph convolutions, both spatial and spectral. Details and proofs are given in Appendix A.

• ChebNet (Defferrard et al., 2016), GAT (Veličković et al., 2017), EdgeNet (Isufi et al., 2020): In ChebNet, $M$ per $(c', c)$ equals a degree-$(L$-1$)$ polynomial of the graph Laplacian matrix, where the polynomial coefficients are trainable. GCN (Kipf & Welling, 2016) can be viewed as ChebNet with polynomial degree-1 and tied coefficients. The attention mechanism in GAT enhances the model expressiveness by incorporating adaptive kernel-based non-negative affinities. In EdgeNet, the graph convolution operator is the product of trainable local filters supported on order-1 neighborhoods. We have the following proposition:

**Proposition 1.** *L3Net* (2) *includes the following models as special cases:*

  *(1) ChebNet (GCN) when $K \geq L$ ($K \geq 2$), $L$ being the polynomial degree.*
  *(2) GAT when $K \geq R$, $R$ being the number of attention branches.*
  *(3) EdgeNet when $K \geq L$, $L$ being the order of graph convolutions.*

• CNN: When nodes lie on a geometrical domain that allows translation $(u' - u)$, in (2) setting $B_k(u', u) = b_k(u' - u)$ for some $b_k(\cdot)$ enforces spatial convolutional. The convolutional kernel can be decomposed as $\sum_k a_k(c', c) b_k(\cdot)$ (Qiu et al., 2018). Extension to CNN on manifold mesh is also possible as in Masci et al. (2015); Fey et al. (2018). We have the following:

**Proposition 2.** *Mesh-based geometrical CNN's defined by linear patch operators, including standard CNN on $\mathbb{R}^d$, and with low-rank decomposed filters are special cases of L3Net* (2).

We also note that L3Net reduces from locally connected GNN (Coates & Ng, 2011; Bruna et al., 2013), the largest class of spatial GNN, only by the low-rankness imposed by a small number of $K$ in (2). Locally connected GNN can be viewed as (1) with the requirement that for each $(c, c')$, $M(u', u; c', c)$ is nonzero only when $u'$ is locally connected in $u$. The complexities of the various models are summarized in Fig. 2 (Table), where L3Net reduces from the $np \cdot CC'$ complexity of locally-connected net to be the additive $(np + CC')$ times $K$.

When the number of channels $C$, $C'$ are large, e.g. in deep layers they $\sim 10^2$, and the graph size is not large, e.g., in landmark data applications $np \ll CC'$, the complexity is dominated by $KCC'$ which is comparable with ChebNet (GAT) if $K \approx L$ ($R$). The computational cost is also comparable, as shown in experiments in Sec. 4. Furthermore, we have:

**Proposition 3.** *Suppose the subgraphs on $N_u^{(d_k)}$ are all connected, given $\alpha_{u,k} > 0$ for all $u, k$, the minimum of (3) with constraint $\|b_u^{(k)}\|_2 \geq \alpha_{u,k}$ is achieved when $b_u^{(k)}$ equals the first Dirichlet eigenvector on $N_u^{(d_k)}$, which does not change sign on $N_u^{(d_k)}$.*

The proposition shows that in the strong regularization limit of $\lambda \to \infty$ in (4), L3Net reduces to be ChebNet-like. The constraint with constants $\alpha_{u,k}$ is included because otherwise the minimizer will be $B_k$ all zero. The first Dirichlet eigenvector is envelope-like (Fig. 2), and then $B_k(\cdot, u)$ will be averaging operators on the local patch. Thus the regularization parameter $\lambda$ can be viewed as trading-off between the more expressiveness in the learnable $B_k$, and the more stability of the averaging local filters, similar to ChebNet and GCN.

## 3    ANALYSIS

We analyze the representation expressiveness and stability (defined in below) of the proposed L3Net model. All proofs in Appendix A, and experimental details in Appendix B.

### 3.1    REPRESENTATION EXPRESSIVENESS OF GRAPH SIGNALS

The theoretical question of graph signal representation expressiveness concerns the ability for GNN deep features to distinguish graph signals. While related, the problem differs from the graph isomorphism test problem which has been intensively studied in the GNN expressiveness literature. Here we prove that L3Net is strictly more expressive than certain spectral GNNs, and support the theoretical prediction by experiments.

We have shown that the L3Net model contains ChebNet (Proposition 1), and the following proposition proves the strictly more expressiveness for graph signal classification. We call $B$ a graph local filter if $B(u, v)$ is non-zero only when $v$ is in the neighborhood of $u$. In a spectral GNN, the graph convolution takes the form as $x \mapsto f(A)x$ where $f$ is a function on $\mathbb{R}$, and $A$ is the (possibly normalized) adjacency matrix.

**Proposition 4.** *There is a graph and 1) A local filter $B$ on it such that $B$ cannot be expressed by any spectral graph convolution, but can be expressed by L3Net with $K = 1$. 2) Two data distributions on the graph (two classes) such that, with a permutation group invariant operator in the last layer, the deep feature of any spectral GNN cannot distinguish the two classes, but that of L3Net with 1 layer and $K = 1$ can.*

The fundamental argument is that spectral GNN is permutation equivariant (see e.g. Gama et al. (2019a), reproduced as Lemma A.1), and the local filters in L3Net break such symmetry to obtain more discriminative power. The constructive example used in the proof is on a ring graph (Fig. A.1, $A$ and the basis $B$), and the two data distributions shown in Fig. 3. Proposition 4 gives that, on the ring graph and using GNN with a global pooling in the last layer, an L3Net layer with $K = 1$ can have classification power while a ChebNet with any order cannot. On a chain graph (removing the connection between two end points in a ring graph), which not exactly follows the theory assumption, since the two graphs only differ at one edge, we expect that it will remain a difficult case for the ChebNet but not for L3Net. To verify the theory, we conduct experiments using a two-layer GNN and the results are in Fig. 3 (table). In the last row, we further impose shared basis across nodes which

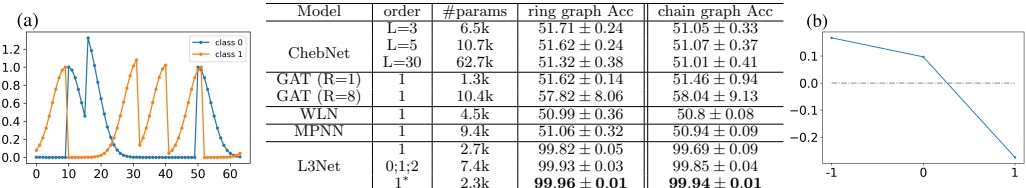

| Model | order | #params | ring graph Acc | chain graph Acc |
|---|---|---|---|---|
| ChebNet | L=3 | 6.5k | $51.71 \pm 0.24$ | $51.05 \pm 0.33$ |
|  | L=5 | 10.7k | $51.62 \pm 0.24$ | $51.07 \pm 0.37$ |
|  | L=30 | 62.7k | $51.32 \pm 0.38$ | $51.01 \pm 0.41$ |
| GAT (R=1) | 1 | 1.3k | $51.62 \pm 0.14$ | $51.46 \pm 0.94$ |
| GAT (R=8) | 1 | 10.4k | $57.82 \pm 8.06$ | $58.04 \pm 9.13$ |
| WLN | 1 | 4.5k | $50.99 \pm 0.36$ | $50.8 \pm 0.08$ |
| MPNN | 1 | 9.4k | $51.06 \pm 0.32$ | $50.94 \pm 0.09$ |
| L3Net | 1 | 2.7k | $99.82 \pm 0.05$ | $99.69 \pm 0.09$ |
|  | 0;1;2 | 7.4k | $99.93 \pm 0.03$ | $99.85 \pm 0.04$ |
|  | 1* | 2.3k | $\mathbf{99.96 \pm 0.01}$ | $\mathbf{99.94 \pm 0.01}$ |

Figure 3: Up/down-wind classification. Plots: (a) Example data from two classes. (b) Learned shared basis on the graph neighborhood of 3, corresponding to the last row in the table. (Table) Test accuracy by MPNN (Gilmer et al., 2017), WLN (Morris et al., 2019), ChebNet up to $L=30$ and L3Net $K=1$ and 3, as well as GAT with different heads. Last row order 1 with star: L3Net with shared basis $B(\cdot, u)$ across all locations $u$.

reduces L3Net to a 1D convolutional layer, and the learned basis shows a "difference" shape (right plot) which explains its classification power. Results are similar using a 1-layer GNN (Tab. A.1). The argument in Proposition 4 extends to other graphs and network types. Generally, when a GNN based on global graph adjacency or Laplacian matrix applies linear combinations of local averaging filters, then certain graph filters may be difficult to express. We experimentally examine GAT, WLN and MPNN, which underperform on the binary classification task, as shown in Fig. 3 (table).

## 3.2 Representation stability

We derive perturbation bounds of GNN feature representation, which is important for robustness against data noise. The analysis implies a trade-off between de-noising and keeping high-frequency information, which is consistent with experimental observation in Sec. 4.

Consider the change in the GNN layer output $Y$ defined in (1)(2) when the input $X$ changes. For simplicity, let $C = C' = 1$, and the argument extends. For any graph signal $x : V \to \mathbb{R}$ and $V' \subset V$, define $\|x\|_{2,V'} := (\sum_{u \in V'} x(u)^2)^{1/2}$ and $\langle x, y \rangle_{V'} = \sum_{u \in V'} x(u)y(u)$. The following perturbation bound holds for the L3Net layer with/without regularization.

**Theorem 1.** *Suppose that $X = \{X(u)\}_{u \in V}$ is perturbed to be $\tilde{X} = X + \Delta X$, the activation function $\sigma : \mathbb{R} \to \mathbb{R}$ is non-expansive, and $\sup_{u \in V} \sum_{k=1}^{K} |N_u^{(d_k)}| \leq Kp$, then the change in the output $\{Y(u)\}_{u \in V}$ in 2-norm is bounded by*

$$\|\Delta Y\|_{2,V} \leq \beta^{(1)} \cdot \|a\|_2 \sqrt{Kp} \|\Delta X\|_{2,V}, \quad \beta^{(1)} := \sup_{k,u} \|B_k(\cdot, u)\|_{2,N_u^{(d_k)}}.$$

Note that $p$ indicates the averaged size of the $d_k$-order local neighborhoods. The proposition implies that when $K$ is $O(1)$, and the local basis $B_k$'s have $O(1)$ 2-norms on all local parches uniformly bounded by $\beta^{(1)}$, then the Lipschitz constant of the GNN layer mapping is $O(1)$, i.e., the product of $\|a\|_2$, $\beta^{(1)}$ and $\sqrt{Kp}$, which does not scale with $n$. This resembles the generalizes the 2-norm of a convolutional operator which only involves the norm of the convolutional kernel, which is possible due to the local receptive fields in the spatial construction of L3Net.

The local graph regularization introduced in Sec. 2.2 improves the stability of $Y$ w.r.t. $\Delta X$ by suppressing the response to local high-frequency perturbations in $\Delta X$. Specifically, the local graph Laplacian $L_u^{(k)}$ on the subgraph on $N_u^{(d_k)}$ is positive definite whenever the subgraph is connected and not isolated from the whole graph. We then define the weighted 2-norm on local patch $\|x\|_{L_u^{(k)}} := \langle x, L_u^{(k)} x \rangle_{N_u^{(d_k)}}$, and similarly $\|x\|_{(L_u^{(k)})^{-1}}$.

**Theorem 2.** *Notation and setting as in Theorem 1, if furtherly, all the subgraphs on $N_u^{(d_k)}$ are connected within itself and to the rest of the graph, and there is $\rho \geq 0$ s.t. $\forall u, k$, $\|\Delta X\|_{(L_u^{(k)})^{-1}} \leq \rho \|\Delta X\|_{2,N_u^{(d_k)}}$, then*

$$\|\Delta Y\|_{2,V} \leq \rho \beta^{(2)} \cdot \|a\|_2 \sqrt{Kp} \|\Delta X\|_{2,V}, \quad \beta^{(2)} := \sup_{k,u} \|B_k(\cdot, u)\|_{L_u^{(k)}}.$$

The bound improves from Theorem 1 when $\rho \beta^{(2)} < \beta^{(1)}$, and regularizing by $\mathcal{R} = \sum_{u,k} \|B_k(\cdot, u)\|_{L_u^{(k)}}^2$ leads to smaller $\beta^{(2)}$. Meanwhile, on each $N_u^{(d_k)}$ the Dirichlet eigenvalues increases $0 < \lambda_1 \leq \lambda_2 \cdots \leq \lambda_{p_{u,k}}$, $p_{u,k} := |N_u^{(d_k)}|$, thus weighting by $\lambda_l^{-1}$ in $\| \cdot \|_{(L_u^{(k)})^{-1}}$ decreases the contribution from high-frequency eigenvectors. As a result, $\rho$ will be small if $\Delta X$ contains a significant high-frequency component on the local patch, e.g., additive Gaussian noise or missing values. Note that in the weighted 2-norm of $\Delta X$ by $(L_u^{(k)})^{-1}$, only the relative amount of high-frequency component in $\Delta X$ matters (because any constant normalization of $L_u^{(k)}$ cancels in the product of $\rho$ and $\beta^{(2)}$). The benefits of local graph regularization in presence of noise in graph data will be shown in experiments.

## 4 Experiment

We test the proposed L3Net model on several datasets [1].

---

[1]Codes available at https://github.com/ZichenMiao/L3Net.

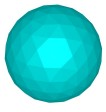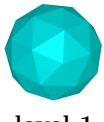

| Model | 4;3;2 Acc | 3;2;1 Acc | 3;2;0 Acc | 3;1;0 Acc | 2;2;1 Acc | 2;1;0 Acc | **3;0;0 Acc** | **2;0;0 Acc** |
|---|---|---|---|---|---|---|---|---|
| UGSCNN | 99.2 | 98.81 | 97.52 | 97.96 | **98.22** | 97.77 | 75.75 | 86.61 |
| GCN | 95.8 | 90.46 | 75.62 | 84.31 | 94.01 | 83.24 | 27.92 | 37.07 |
| ChebNet | **99.3** | 98.50 | 98.07 | 97.07 | 97.12 | 95.51 | 73.1 | 90.73 |
| **L3Net** (1;1;2;3) | 99.1 | **98.81** | **98.89** | **98.60** | 97.76 | **97.97** | **93.14** | **97.26** |

**level 2**      **level 1**

Figure 4: (Plot) Icosahedral spherical meshes at level 2 and 1. (Table) Testing accuracies of sphere MNIST under different mesh settings, $(l1; l2; l3)$ stands for the mesh level used in each GNN layer. L3Net uses $K=4$, and neighborhood order (1;1;2;3). S2CNN (Cohen et al., 2018) on mesh (4;3;2) has accuracy 96.0.

### 4.1 OBJECT RECOGNITION OF DATA ON SPHERICAL MESH

We first classify data on a spherical mesh: sphere MNIST and sphere ModelNet-40, following the settings in literature. Though regular mesh on sphere is not the primary application scenario that motivates our model, we include the experiments to compare with benchmarks and test the efficiency of L3Net on such regular meshes. Following UGSCNN (Jiang et al., 2019), we implement different mesh resolution on a sphere, indicated by "mesh level" (Fig. 4), where number of nodes in different levels can vary from 2562 (level 4) to 12 (level 0). All the networks consist of three convolutional layers, see more details in Appendix C.1. Using the original mesh level (4;3;2), the finest resolution as in UGSCNN, L3Net gives among the best accuracies for sphere MNIST. On Modelnet-40, L3Net achieves a testing accuracy of 90.24, outperforming ChebNet and GCN and and is comparable to UGSCNN which uses spherical mesh information (Tab. A.2). When the mesh becomes coarser, as shown in Fig. 4 (Table), L3Net improves over GCN and ChebNet ($L=4$) and is comparable with UGSCNN under nearly all mesh settings. We observe that in some settings ChebNet can benefit from larger $L$, but the overall accuracy is still inferior to L3Net. The most right two columns give two cases of coarse meshes where L3Net shows the most significant advantage.

### 4.2 FACIAL EXPRESSION RECOGNITION (FER)

We test on two FER datasets, Extended CohnKanade (CK+) (Lucey et al., 2010) and FER13 (Goodfellow et al., 2013). We use 15 facial landmarks, see Fig. 1, and pixel values on a patch around each landmark point as node features. Details about dataset and model setup are in Appendix C.2. Unlike spherical mesh, facial and body landmarks are coarse irregular grids where no clear pre-defined mesh operation is applicable. We benchmark L3Net with other GNN approaches, as shown in Table 1. The local graph regularization strategy is applied on FER13, due to the severe outlier data of landmark detection caused by occlusion. On CK+, L3Net leads all non-CNN models by a large margin, and the best model (1,1,2,3) uses comparable number of parameters with the best ChebNet ($L=4$). On FER13, L3Net has lower performance than ChebNet and EdgeNet (Isufi et al., 2020), but outperforms after adding regularization. The running times of best ChebNet and L3Net models are comparable, and are much less than GAT's.

Table 1: Results on CK+ and FER13, with comparison to CNN[†](Ding et al., 2017), CNN[‡] (Guo et al., 2016), landmark method using handcrafted features (Morales-Vargas et al., 2019), and various GNN methods. Specifically, we compare to GAT (Veličković et al., 2017) with different #heads (h) and #features (f). The mean testing time on CK+: ChebNet ($L=4$) 12.56ms, L3Net (order 1,1,2,3) 13.02ms. GAT (h=f=8) 39.67ms, (h=f=16) 41.02ms.

| Model | Bases Order | CK+ #params (w/o FC) | CK+ Acc | FER13 #params (w/o FC) | FER13 Acc |
|---|---|---|---|---|---|
| CNN[†] | - | 7M | 98.60 | - | - |
| CNN[‡]. | - | - | - | 2.6M | 71.33 |
| Landmarks-handcraft | - | - | $91.00 \pm 0.03$ | - | - |
| GAT (h=8, f=8) | 1 | 34.6k | $91.62 \pm 1.16$ | 46.9k | 49.50 |
| GAT (h=16, f=16) | 1 | 142.3k | $90.87 \pm 0.78$ | 151.1k | 48.93 |
| GCN | 1 | 34.5k | $91.78 \pm 0.38$ | 42.6k | 55.54 |
| GraphConv | 1 | 169.6k | $81.62 \pm 0.48$ | 215.4k | 55.63 |
| ChebNet | $L=3$ | 102.3k | $92.93 \pm 0.59$ | 136.4k | 59.68 |
| | $L=4$ | 136.3k | $93.22 \pm 0.37$ | 181.6k | 60.26 |
| | $L=5$ | 170.2k | $93.03 \pm 0.62$ | 227.3k | 60.29 |
| EdgeNet | $L=3$ | 103.4k | $92.41 \pm 0.81$ | 137.2k | 58.73 |
| | $L=4$ | 137.1k | $92.57 \pm 0.84$ | 182.5k | 60.05 |
| **L3Net** | 2;2;2 | 102.8k | $95.32 \pm 0.31$ | 139.7k | 60.46 |
| | 0;1;2;3 | 136.8k | $95.03 \pm 0.30$ | 182.8k | 60.65 |
| | 1;1;2 +reg0.005 | 102.7k | $94.68 \pm 0.56$ / $94.52 \pm 0.61$ | 139.4k | 59.68 / 61.13 |
| | 1;1;2;3 +reg0.5 | 136.9k | $\mathbf{95.37 \pm 0.60}$ / $95.11 \pm 0.44$ | 183.0k | 60.71 / **61.64** |

Table 2: Results on NTU-RGB+D and Kinetics-Motion

| Model | Bases order | NTU-RGB+D | | | Kinetics-Motion | |
| | | #params (w/o FC) | x-view Acc | x-sub Acc | #params (w/o FC) | Acc |
|---|---|---|---|---|---|---|
| ST-GCN (Yan et al., 2018) | 1 | - | 88.30 | 81.50 | - | 72.4 |
| ST-GCN | 1 | 2.6M | 82.59 | 74.33 | 1.4M | 72.85 |
| ST-ChebNet | $L$=3 | 3.1M | 86.40 | 78.24 | 1.8M | 77.91 |
| | $L$=4 | 3.3M | 86.45 | 80.20 | 2.1M | 78.24 |
| | $L$=5 | 3.5M | 76.70 | 71.42 | 2.3M | 77.57 |
| **ST-L3Net** | 1;1;2 +reg0.01 | 3.1M | 90.78 / 88.38 | **83.64** / 81.54 | 1.8M | 75.20 / **78.49** |
| | 1;1;2;3 +reg0.01 | 3.3M | **91.52** / 89.87 | 82.46 / 80.97 | 2.1M | 75.07 / 76.68 |

## 4.3 ACTION RECOGNITION

We test on two skeleton-based action recognition datasets, NTU-RGB+D (Shahroudy et al., 2016) and Kinetics-Motion (Kay et al., 2017). The irregular mesh is the 18/25-point body landmarks, with graph edges defined by body joints, shown in Fig. 1 and Fig. A.2. We adopt ST-GCN (Yan et al., 2018) as the base architecture, and substitute the GCN layer with new L3Net layer, called **ST-L3Net**. On Kinetics-Motion, we adopt the regularization mechanism to overcome the severe data missing caused by camera out-of-view. See more experimental details in Appendix C.3. We benchmark performance with ST-GCN (Yan et al., 2018), ST-GCN (our implementation without using geometric information) and ST-ChebNet (replacing GCN with ChebNet layer), shown in Table 2. L3Net shows significant advantages on two NTU tasks, cross-view and cross-subject settings. On Kinetics-Motion, L3Net regains superiority over other models after applying regularization. The results in both Table 1 and 2 indicate that stronger regularization sacrifices expressiveness for clean data and gains stability for noisy data, which is consistent with the theory in Sec. 3.2.

## 4.4 ROBUSTNESS TO GRAPH NOISE

To examine the robustness to graph noise, we experiment on down-sampled MNIST data on 2D regular grid with 4-nearest-neighbor graph. With no noise, on 28×28 data (Tab. A.3), 14×14 data (Tab. A.4), and 7×7 data (Tab. 3 "original" column), the performance of L3Net is comparable to ChebNet (Defferrard et al., 2016) and EdgeNet (Isufi et al., 2020) and better than other GNN methods. We consider three types of noise, Gaussian noise added to the pixel value, missing nodes or equivalently missing value in image input, and permutation of the node indices, details in Appendix C.4. The results of adding different levels of gaussian noise and permutation noise are shown in Tab. 3, while results of adding missing value noise is provided in Appendix C.4. The results show that our regularization scheme improves the robustness to all three types of graph noise, supporting the theory in Sec. 3.2. Specifically, L3Net without regularization may underperform than ChebNet, but catches up after adding regularization, which is consistent with Proposition 3.

Table 3: Results on MNIST with grid size $7 \times 7$ with different levels of Gaussian noise and Permutation noise.

| Model | bases order | #params (w/o FC) | Acc(original) | Acc (gaussian) (psnr 24.9) | Acc (gaussian) (psnr 19.1) | Acc (gaussian) (psnr 15.7) | Acc (permutation) |
|---|---|---|---|---|---|---|---|
| GCN | 1 | 2.4k | $90.02 \pm 0.24$ | $89.27 \pm 0.09$ | $85.70 \pm 0.13$ | $81.32 \pm 0.18$ | $83.00 \pm 0.18$ |
| ChebNet | $L$=3 | 6.5k | $92.85 \pm 0.09$ | $91.13 \pm 0.15$ | $87.64 \pm 0.23$ | $82.70 \pm 0.33$ | $86.94 \pm 0.06$ |
| | $L$=5 | 10.7k | $93.2 \pm 0.07$ | $91.92 \pm 0.11$ | $88.22 \pm 0.10$ | $83.04 \pm 0.12$ | $87.27 \pm 0.23$ |
| | $L$=7 | 14.8k | $93.45 \pm 0.06$ | $91.80 \pm 0.10$ | $87.84 \pm 0.15$ | $83.75 \pm 0.14$ | $87.53 \pm 0.19$ |
| GAT (h=8,f=16) | 1 | 17.5k | $79.50 \pm 1.24$ | $68.68 \pm 0.45$ | $64.8 \pm 1.69$ | $65.38 \pm 1.03$ | $62.21 \pm 0.56$ |
| MPNN | 1 | 18.8k | $86.94 \pm 0.37$ | $85.36 \pm 0.51$ | $82.23 \pm 0.35$ | $77.59 \pm 0.34$ | $77.55 \pm 0.26$ |
| WLN | 1 | 17.1k | $87.61 \pm 0.04$ | $86.01 \pm 0.20$ | $83.60 \pm 0.09$ | $79.47 \pm 0.11$ | $80.51 \pm 0.05$ |
| EdgeNet | $L$=3 | 7.5k | $93.26 \pm 0.16$ | $91.81 \pm 0.14$ | $88.42 \pm 0.36$ | $84.56 \pm 0.40$ | $87.15 \pm 0.30$ |
| | $L$=4 | 10.1k | $93.44 \pm 0.17$ | $92.27 \pm 0.16$ | $88.60 \pm 0.17$ | $84.15 \pm 0.59$ | $87.44 \pm 0.28$ |
| **L3Net** | 0;1;2 | 8.1k | $93.45 \pm 0.10$ | - | - | - | - |
| | 1;1;2 +reg0.5 | 8.4k | $93.56 \pm 0.08$ / **$93.85 \pm 0.13$** | $92.10 \pm 0.08$ / $92.31 \pm 0.07$ | $88.20 \pm 0.13$ / **$89.23 \pm 0.10$** | $83.00 \pm 0.33$ / $84.59 \pm 0.23$ | $87.58 \pm 0.19$ / $88.08 \pm 0.18$ |
| | 1;1;2;3 +reg0.5 | 12.2k | $93.67 \pm 0.15$ / $93.85 \pm 0.15$ | $92.25 \pm 0.15$ / **$92.56 \pm 0.12$** | $88.28 \pm 0.16$ / $89.15 \pm 0.24$ | $82.80 \pm 0.37$ / **$84.61 \pm 0.25$** | $87.66 \pm 0.12$ / **$88.21 \pm 0.15$** |

## 5 CONCLUSION AND DISCUSSION

The paper proposes a new graph convolution model using learnable local filters decomposed over a small number of basis. Strengths: Provable enhancement of model expressiveness with significantly reduced model complexity from locally connected GNN. Improved stability and robustness via local graph regularization, supported by theory. Plug-and-play layer type, suitable for GNN graph signal classification problems on relatively unchang-

ing small underlying graphs, like face/body landmark data in FER and action recognition applications.

Limitations and extensions: (1) Scalability to larger graph. When $|V| = n$ is large, the complexity increase in the $npK$ term would be significant. The issue in practice can be remedied by mixing use of layer types, e.g., only adopting L3Net layers in upper levels of mesh which are of reduced size. (2) Dynamically changing underlying graph across samples. For more severe changes of the underlying graph, we can benefit from solutions such as node registration or other preprocessing techniques, possibly by another neural network. Related is the question of reducing model dependence on graph topology, possibly under a statistical model of the underlying graphs. This includes transferability to larger networks. (3) Incorporation of edge features. Edge features can be transformed into extra channels of node features by an additional layer in the bottom, and the low-rank graph operation can be similarly employed there. (4) Theoretically, the representation robustness analysis is to be extended to more general types of graph perturbation. Generally, one can work to extend to other types of graph data and tasks.

## ACKNOWLEDGEMENTS

The work is supported by NSF (DMS-1820827). XC is partially supported by NIH and the Alfred P. Sloan Foundation. ZM and QQ are partially supported by NSF and the DARPA TAMI program.

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

## Appendix

## A  Proofs

### A.1  Details and proofs in Sec. 2.3

To facilitate comparison with literature, we provide a summary of various graph convolution models in matrix notation, the precise definition of which will be detailed in below. For simplicity, only the linear transform part is shown, and the addition of bias and point-wise non-linearity are omitted.

Notation as in Section 2.1, suppose $X \in \mathbb{R}^{n \times C'}$ is the input node feature, and $Y \in \mathbb{R}^{n \times C}$ the output feature,

- L3Net (ours): $Y = \sum_{k=1}^{K} B_k X A_k$, where $B_k \in \mathbb{R}^{n \times n}$ is the local basis filter, and $A_k \in \mathbb{R}^{C' \times C}$ are the coefficients, both $B_k$ and $A_k$ are learnable.
- ChebNet/GCN: $Y = \sum_{l=0}^{L-1} T_l(\tilde{L}) X \Theta_l$, where $T_l(\cdot)$'s are Chebshev polynomials, $\tilde{L}$ is the rescaled and re-centered graph Laplacian, $T_l(\tilde{L}) \in \mathbb{R}^{n \times n}$, and $\Theta_l \in \mathbb{R}^{C' \times C}$ are trainable.
- GAT: $Y = \sum_{r=1}^{R} \mathcal{A}^{(r)} X \Theta_r$, where $\mathcal{A}^{(r)} \in \mathbb{R}^{n \times n}$ is the graph attention affinity computed adaptively from input features, and $\Theta_r \in \mathbb{R}^{C' \times C}$ are trainable and weight-shared with the parameters in $\mathcal{A}^{(r)}$, see more in below.
- EdgeNet: $Y = \sum_{r=0}^{L-1} P_r X \Theta_r$, where $P_r = \prod_{k=0}^{r} \Phi_k$ for a sequence of trainable local filters $\Phi_k$, and $\Theta_r \in \mathbb{R}^{C' \times C}$ are trainable.

From the matrix formulation, it can be seen that when $B_k$ are the classical graph filtering operators, e.g. polynomials of $\tilde{L}$, and $A_k$ the trainable $\Theta_k$, L3Net recovers the above graph convolution models in literature (c.f. Proposition 1). In below we give more details, as well as the reduction to filter-decomposed CNN (c.f. Proposition 2).

#### A.1.1  Locally connected GNN

Specifically, the construction in Coates & Ng (2011); Bruna et al. (2013) assumes that $u$ and $u'$ belongs to the graph of different scales, $u'$ is on the fine graph, and $u$ is on a coarse-grained layer produced by clustering of indices of the graph of the input layer. If one generalize the construction to allow over-lapping of the receptive fields, and assume no pooling or coarse-graining of the graph, then the non-zero parameters are of the number

$$\sum_{u \in V} |N_u| \cdot CC' = np \cdot CC',$$

where $n = |V|$, $p$ is the average patch size $|N_u|$, and $C$ and $C'$ are the number of input and output feature channels.

A.1.2   CHEBNET/GCN, GAT AND EDGENET

● ChebNet/GCN

In view of (1), ChebNet (Defferrard et al., 2016) makes use of the graph adjacency matrix to construct $M$. Specifically, $A_{sym} := D^{-1/2}AD^{-1/2}$ is the symmetrized graph adjacency matrix (possibly including self-edge, then $A$ equals original $A$ plus $I$), and $L_{sym} := I - A_{sym}$ has spectral decomposition $L_{sym} = \Psi\Lambda\Psi^T$. Let $\tilde{L} = \alpha_1 I + \alpha_2 L_{sym}$ be the rescaled and re-centered graph Laplacian such that the eigenvalues are between $[-1, 1]$, $\alpha_1, \alpha_2$ fixed constants. Then, written in $n$-by-$n$ matrix form,

$$M_{c',c} = \sum_{l=0}^{L-1} \theta_l(c', c) T_l(\tilde{L}), \quad \theta_l(c', c) \in \mathbb{R}, \tag{5}$$

where $T_l(\cdot)$ is Chebshev polynomial of degree $l$. As $A_{sym}$ and then $\tilde{L}$ are given by the graph, only $\theta_l$'s are trainable, thus the number of parameters are

$$L \cdot CC'.$$

GCN (Kipf & Welling, 2016) is a special case of ChebNet. Take $L = 2$ in (5), and tie the choice of $\theta_0$ and $\theta_1$,

$$M_{c',c} = \theta(c', c)(\alpha_1' I + \alpha_2' A_{sym}) =: \theta(c', c)\tilde{A}, \quad \alpha_1', \alpha_2' \text{ fixed constants,}$$

where $\theta(c', c)$ is trainable. This factorized form leads to the linear part of the layer-wise mapping as $Y = \tilde{A}X\Theta$ written in matrix form, where $\tilde{A}$ is $n$-by-$n$ matrix defined as above, $X$ ($Y$) is $n$-by-$C'$ ($-C$) array, $\Theta$ is $C'$-by-$C$ matrix. The model complexity is $CC'$ which are the parameters in $\Theta$.

● GAT

In GAT (Veličković et al., 2017), $R$ being the number of attention heads, the graph convolution operator in one GNN layer can be written as (omitting bias and non-linear mapping)

$$Y = \sum_{r=1}^{R} \mathcal{A}^{(r)} X\Theta_r, \quad \mathcal{A}_{u,v}^{(r)} = \frac{e^{c_{uv}^{(r)}}}{\sum_{v' \in N_u^{(1)}} e^{c_{uv'}^{(r)}}}, \quad c_{uv}^{(r)} = \sigma((a^{(r)})^T [W^{(r)} X_u, W^{(r)} X_v]), \tag{6}$$

where $\{W^{(r)}, a^{(r)}\}$ are the trainable parametrization of attention graph affinity mechanism $\mathcal{A}^{(r)}$, which constructs non-negative affinities between graph nodes $u$ and $v$ adaptively from the input graph node feature $X$. In particular, $\mathcal{A}^{(r)}$ shares sparsity pattern as the graph topology, that is, $\mathcal{A}^{(r)}(u, u') \neq 0$ only when $u' \in N_u^{(1)}$.

In the original GAT, $\Theta_r = W^{(r)}\mathbf{C}^{(r)}$, where $\mathbf{C}^{(r)}$'s are fixed matrices such that the output from $r$-th head is concatenated into the output $Y$ across $r = 1, \cdots, R$. Variants of GAT adopt channel mixing across heads, e.g. a generalization of GAT in Isufi et al. (2020) uses extra trainable $\Theta_r$ in (6) independent from $W^{(k)}$. Isufi et al. (2020) also proposed higher-order GAT by considering powers of the affinity matrix $\mathcal{A}^{(r)}$ as well as the edge-varying version (c.f. Eqn. (36)(39) in Isufi et al. (2020)). As this higher-order GAT and the edge-varying counterpart are special cases of the edgy-varying GNN, we cover this case in Proposition 1 3).

The model complexity of GAT: In the original GAT where $\Theta_r$ is tied with $W^{(r)}$, the number of parameters in one layer is $R(C_0 C' + 2C_0)$, where $R$ is the number of attention heads, $C = C_0 R$, and $W^{(r)} : \mathbb{R}^{C'} \to \mathbb{R}^{C_0}$. When $\Theta_r$ are free from $\{W^{(r)}, a^{(r)}\}$ in (6), the number of parameters is $R(CC' + C_0 C' + 2C_0) \leq R(2CC' + 2C)$, where $W^{(r)}$ maps to dimension $C_0$ and $\Theta_r$ maps to dimension $C$.

● EdgeNet (Edge-varying GCN)

Per Eqn. (1)(8) in Isufi et al. (2020), the edge-varying GNN layer mapping can be written as

$$Y = \sum_{r=0}^{L-1} \left( \prod_{k=0}^{r} \Phi_k \right) X\Theta_r, \tag{7}$$

where $\Phi_0$ is an $n$-by-$n$ diagonal matrix, and $\Phi_k$, $k = 1, \cdots, r$, are supported on $N_u^{(1)}$ of each node $u$. The trainable parameters are $\{\Phi_k\}_{k=0}^R$ and $\{\Theta_r\}_{r=0}^R$, $\Theta_r : \mathbb{R}^{C'} \to \mathbb{R}^C$. Edge-varying GAT implements polynomials of averaging filters, and general edge-varying GNN takes product of arbitrary 1-order filters. The proof shows that EdgeNet layer is a special case of L3Net layer, while restricting $B_k$ to be of the product form (9) rather than freely supported on $N_u^{(d_k)}$ for user-specified order $(d_1, \cdots, d_K)$ is a non-trivial restriction.

The trainable parameters: $\Theta_r$ has $LCC'$ many, $\Phi_0$ has $n$, and $\Phi_k$, $k = 1, \cdots, L-1$ each has $np^{(1)}$ many, $p^{(1)}$ being the average size o 1-neighborhood of nodes. Thus the total number of parameters is

$$LCC' + n + (L-1)np^{(1)} \sim L(CC' + np^{(1)}).$$

*Proof of Proposition 1.* Part (1): Since GCN is a special case of ChebNet, it suffices to prove that (5) can be expressed in the form of L3Net (2) for some $K$. By definition of $\tilde{L}$, mathematically equivalently,

$$M_{c',c} = \sum_{l=0}^{L-1} \theta_l(c', c) T_l(\alpha_1 I + \alpha_2 L) = \sum_{l=0}^{L-1} \theta_l(c', c) T_l(\alpha_1 I + \alpha_2 (I - A_{sym})) = \sum_{l=0}^{L-1} \beta_l(c', c) A_{sym}^l, \tag{8}$$

where the coefficients $\beta_l$'s are determined by $\theta_l$'s, per $(c', c)$. Since $A_{sym}^l$ propagates to the $l$-th order neighborhood of any node, setting $B_k(u', u) = A_{sym}^{k-1}(u', u)$, $B_k(u', u)$ is non-zero when $u' \in N_u^{(k-1)}$, $1 \le k \le K := L$, and then setting $a_k(c', c) = \beta_{k-1}(c', c)$ gives (5) in the form of (2).

Part (2): We consider (6) as the GAT model. Recall that $\Theta_r : \mathbb{R}^{C'} \to \mathbb{R}^C$, then (6) can be re-written in the form of (1) by letting

$$M(u', u; c', c) = \sum_{r=1}^R \mathcal{A}^{(r)}(u', u) \Theta_r(c', c),$$

which is a special case of (2) where $R = K$, $\mathcal{A}^{(k)} = B_k$ and $\Theta_k = a_k$. Since $\mathcal{A}^{(r)}(u, u')$ as a function of $u'$ is supported on $u' \in N_u^{(1)}$, (6) belongs to the L3Net model (2) where $d_1 = \cdots = d_K = 1$, in addition to that $B_k$ must be of the attention affinity form, i.e. built from the attention coefficients $c_{uv}^{(r)}$ computed from input $X$ via parameters $\{W^{(r)}, a^{(r)}\}$.

Part (3): Comparing with (1)(2), we have that (7) is a special case of L3Net (2) by letting $K = L$,

$$B_k = \prod_{k'=0}^{k-1} \Phi_{k'}, \tag{9}$$

$a_k = \Theta_{k-1}$, and $d_k = k - 1$ for $k = 1, \cdots, K$. □

### A.1.3 Standard and geometrical CNN's

Standard CNN on $\mathbb{R}^d$, e.g. $d = 1$ for audio signal and $d = 2$ for image data, applies a discretized convolution to the input data in each convolutional layer, which can be written as (omitting bias which is added per $c$, and the non-linear activation)

$$y(u, c) = \sum_{c' \in [C']} \sum_{u' \in U} w_{c',c}(u' - u) x(u', c'), \tag{10}$$

where $U$ is a grid on $\mathbb{R}^d$. We write in the way of "anti-convolution", which has "$u' - u$" rather than "$u - u'$", but the definition is equivalent. For audio and image data, $U$ is usually a regular mesh with evenly sampled grid points, and proper boundary conditions are applied when computing $y(u, c)$ at a boundary grid point $u$. E.g., boundary can be handled by standard padding as in CNN. As the convolutional filters $w_{c',c}$ are compactly supported, the summation of $u'$ is on a neighborhood of $u$.

More generally, CNN's on non-Euclidean domains are constructed when spatial points are sampled on an irregular mesh in $\mathbb{R}^d$, e.g., a 2D surface in $\mathbb{R}^3$. The generalization of (10) is by defining the "patch operator" (Masci et al., 2015) which pushes a template filter $w$ on a regular mesh on $\mathbb{R}^d$, $d$ being the intrinsic dimensionality of the sampling domain, to the irregular mesh in the ambient space that have coordinates on local charts. Specifically, for a mesh of 2D surface in 3D, $d = 2$, and $w$ is a template convolutional filter on $\mathbb{R}^2$. For any local cluster of 3D mesh points $N_u$ around a point $u$, the patch operator $\mathcal{P}_u$ provides $(\mathcal{P}_u w)(u')$ for $u' \in N_u$ by certain interpolation scheme on the local chart. The operator $\mathcal{P}_u$ is linear in $w$, and possibly trainable. As a result, in mesh-based geometrical CNN,

$$y(u, c) = \sum_{c' \in [C']} \sum_{u'} (\mathcal{P}_u w_{c',c})(u') x(u', c'), \tag{11}$$

and one can see that in Euclidean space taking $(\mathcal{P}_u w)(u') = w(u' - u)$ reduces (11) to the standard CNN as in (10).

In both (10) and (11), spatial low-rank decomposition of the filters $w_{c',c}$ can be imposed (Qiu et al., 2018). This introduces a set of bases $\{b_k\}_k$ over space that linearly span the filters $w_{c',c}$. For standard CNN in $\mathbb{R}^d$, $b_k$ are basis filters on $\mathbb{R}^d$, and for geometrical CNN, they are defined on the reference domain in $\mathbb{R}^d$ same as $w_{c',c}$, where $d$ is the intrinsic dimension. Suppose $w_{c',c} = \sum_{k=1}^K \beta_{k,(c',c)} b_k$ for coefficients $\beta_{k,(c',c)}$, by linearity, (11) becomes

$$y(u, c) = \sum_{c' \in [C']} \sum_{u'} \sum_{k=1}^K \beta_{k,(c',c)} (\mathcal{P}_u b_k)(u') x(u', c'), \tag{12}$$

and similarly for (10). The trainable parameters in (12) are $\beta_{k,(c',c)}$ and the basis filters $b_k$'s, the former has $KCC'$ parameters, and the latter has $\sum_k p_k$, where $p_k$ is the size of the support of $b_k$ in $\mathbb{R}^d$. Suppose the average size is $p$, then the number of parameters is $Kp$. This gives the total number of parameters as

$$KCC' + Kp.$$

*Proof of Proposition 2.* Since standard CNN is a special case of geometrical CNN 11, we only consider the latter. Assuming low-rank filter decomposition, the convolutional mapping is (12). Comparing to the GNN layer mapping defined in (1), one sees that

$$M(u', u; c', c) = \sum_{k=1}^K \beta_{k,(c',c)} (\mathcal{P}_u b_k)(u'),$$

which equals (2) if setting $B_k(u', u) = (\mathcal{P}_u b_k)(u')$ and $a_k(c', c) = \beta_{k,(c',c)}$. □

### A.1.4 STRONG REGULARIZATION LIMIT

*Proof of Proposition 3.* The constrained minimization of $\mathcal{R}$ defined in (3) separates for each $u, k$, and the minimization of $b_u^{(k)}$ is given by

$$\min_{w: N_u^{(d_k)} \to \mathbb{R}} w^T L_u^{(k)} w, \quad \text{s.t. } \|w\|_2 \geq \alpha_{u,k} > 0. \tag{13}$$

For each $u, k$, the local Dirichlet graph Laplacian $L_u^{(k)}$ has eigen-decomposition $L_u^{(k)} = \Psi_u^{(k)} \Lambda_u^{(k)} (\Psi_u^{(k)})^T$, where $(\Psi_u^{(k)})^T \Psi_u^{(k)} = I$, and the diagonal entries of $\Lambda_u^{(k)}$ are eigenvalues of $L_u^{(k)}$, which are all $\geq 0$ and sorted in increasing order. By the variational property of eigenvalues, the minimizer of $w$ in (13) is achieved when $w = \Psi_u^{(k)}(\cdot, 1)$, i.e., the eigenvector associated with the smallest eigenvalue of $L_u^{(k)}$. By that the local subgraph is connected, this smallest eigenvalue has single multiplicity, and the eigenvector is the Perron-Frobenius vector which does not change sign. The claim holds for arbitrary $\alpha_{u,k} > 0$ since eigenvector is defined up to a constant multiplication. □

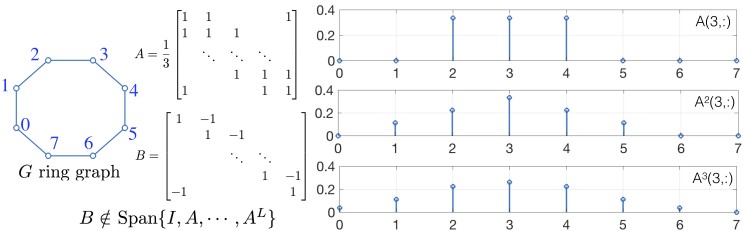

Figure A.1: A ring graph with 8 nodes. Polynomials of graph adjacency matrix $A$ (or Laplacian matrix) preserve symmetry of mirroring around any node, e.g., node 3, and can cannot express a local filter $B$

## A.2 Proofs in Sec. 3.1

*Proof of Proposition 4.* Part 1): Let the graph be the ring graph with $n$ nodes, and each node has 2 neighbors, $n=8$ as shown in Fig. 1 (right). We index the nodes as $u = 0, \ldots, n-1$ and allows addition/subtraction of $u-v \pmod n$. Let $B$ be the "difference" filter $B(u', u) = 1$ when $u' = u$ and $-1$ when $u' = u+1$. We show that $B \neq f(A)$ for any $f$, and in contrast, setting this $B$ as the basis in (2) expresses the filter with $K = 1$.

To prove that $B \neq f(A)$ for any $f$, let $\pi_u$ be the permutation of the $n$ nodes such that $\pi_u(u + v) = (u - v)$ for all $v$, i.e., mirror flip the ring around the node $u$. By construction, the graph topology of the ring graph is preserved under $\pi_u$, that is, $A_{\pi_u} := \pi_u A \pi_u^T = A$, whether $A$ is the 0/1 value adjacency matrix or the symmetrically normalized one $A_{sym} = D^{-1/2} A D^{-1/2}$ ($D$ is constant on diagonal) or other normalized version as long as the relation $A_{\pi_u} = A$ holds. By Lemma A.1 1), for any $f : \mathbb{R} \to \mathbb{R}$,

$$f(A)\pi_u = f(A_{\pi_u})\pi_u = \pi_u f(A),$$

this means that if $B = f(A)$ for some $f$, then $B\pi_u = \pi_u B$, which contradicts with the construction of $B$.

Part 2): Consider the two distributions of graph signals on the ring graph in 1), which we call "upwind/downwind" signals: $X_{up}$ consists of finite superpositions of functions on the ring graph which are periodic, smoothly increasing from 0 to 1 and then dropping to zero. Signals in $X_{up}$ are under certain distribution, and $X_{down}$ consists of the signals that can be produced by mirror-flipping the upwind signals. That is, denoting $x_{up}$ ($x_{down}$) an upwind (downwind) signal, $\pi_u$ the permutation as in 1) around any node $u$, then

$$\pi_u x_{up} \overset{\text{dist.}}{=} x_{down},$$

where $\overset{\text{dist.}}{=}$ means equaling in distribution. Example signals of the two classes as illustrated in Fig. 3.

Same as in 1), by construction $A_{\pi_u} = A$. Let $F^{(L)}$ be the mapping to the $L$-th layer spectral GNN feature, for $x_{up}$ an upwind signal, Lemma A.1 2) gives that

$$F^{(L)}[A]\pi_u x_{up} = F^{(L)}[A_{\pi_u}]\pi_u x_{up} = \pi_u F^{(L)}[A]x_{up}.$$

The last layer applies group invariant operator $U$, then

$$UF^{(L)}[A]\pi_u x_{up} = U\pi_u F^{(L)}[A]x_{up} = UF^{(L)}[A]x_{up},$$

this gives that

$$UF^{(L)}[A]x_{down} \overset{\text{dist.}}{=} UF^{(L)}[A]\pi_u x_{up} = UF^{(L)}[A]x_{up},$$

which means that the final output deep feature via $UF^{(L)}[A]$ are statistically the same for the input signals from the two classes.

Meanwhile, the difference local filter $B$ in the proof of 1) can extract feature to differentiate the two classes: with Relu activation function, the output feature after one convolutional layer and a global pooling, which is permutation invariant, can be made strictly positive for one class, and zero for the other class. Thus, L3Net with 1 layer and 1 basis suffices to distinguish the $X_{up}$ and $X_{down}$ signals. $\qquad \square$

**Lemma A.1** (Permutation equivariance, Proposition 1 in Gama et al. (2019a)). *Let $A$ be the (possibly normalized) graph adjacency matrix, for any input signal $x : V \to \mathbb{R}$, and $\pi \in \mathcal{S}_n$ a permutation of graph nodes,*

*1) The spectral graph convolution mapping $f(A)$ satisfies that*

$$f(A_\pi)\pi = \pi f(A), \quad A_\pi := \pi A \pi^T.$$

*2) Let $F^{(l)}[A]$ be the mapping to the l-th layer spectral GNN feature with graph adjacency $A$, then*

$$F^{(l)}[A_\pi]\pi x = \pi F^{(l)}[A]x.$$

*Proof of Lemma A.1.* Proved in Gama et al. (2019a) and we reproduce with our notation for completeness.

Part 1): Denote the $n$-by-$n$ permutation matrix also by $\pi$, then by definition, $f(A) = Uf(\Lambda)U^T$ where $A = U\Lambda U^T$ is the diagonalization and $U$ is orthogonal matrix, thus

$$f(A_\pi) = f(\pi U \Lambda U^T \pi^T) = \pi U f(\Lambda) U^T \pi^T = \pi f(A) \pi^T,$$

and this proves 1).

Part 2): Each spectral GNN layer mapping adds the bias and the node-wise non-linear activation mapping to the graph convolution linear operator, which preserves the permutation equivariance. Recursively applying to $L$ layers proves 2). □

### A.3 Proofs in Sec. 3.2

*Proof of Theorem 1.* By definition,

$$Y(u) = \sigma\left(\sum_{k=1}^{K} a_k \langle B_k(\cdot, u), X(\cdot)\rangle_{N_u^{(d_k)}} + \text{bias}\right),$$

then since $\sigma$ is non-expansive, $\forall u \in V$,

$$|\Delta Y(u)| \le |\sum_{k=1}^{K} a_k \langle B_k(\cdot, u), \Delta X(\cdot)\rangle_{N_u^{(d_k)}}| \le \|a\|_2 \left(\sum_{k=1}^{K} |\langle B_k(\cdot, u), \Delta X(\cdot)\rangle_{N_u^{(d_k)}}|^2\right)^{1/2}. \quad (14)$$

By that

$$|\langle B_k(\cdot, u), \Delta X(\cdot)\rangle_{N_u^{(d_k)}}| \le \|B_k(\cdot, u)\|_{2, N_u^{(d_k)}} \cdot \|\Delta X(\cdot)\|_{2, N_u^{(d_k)}}, \quad (15)$$

we have that

$$\sum_{u \in V} |\Delta Y(u)|^2 \le \|a\|_2^2 \sum_u \sum_{k=1}^{K} |\langle B_k(\cdot, u), \Delta X(\cdot)\rangle_{N_u^{(d_k)}}|^2$$

$$\le \|a\|_2^2 \sum_u \sum_{k=1}^{K} \|B_k(\cdot, u)\|_{2, N_u^{(d_k)}}^2 \cdot \|\Delta X(\cdot)\|_{2, N_u^{(d_k)}}^2$$

$$\le (\|a\|_2 \beta^{(1)})^2 \sum_{u,k} \|\Delta X(\cdot)\|_{2, N_u^{(d_k)}}^2, \quad (16)$$

and observe that

$$\sum_{u,k} \|\Delta X(\cdot)\|_{2, N_u^{(d_k)}}^2 = \sum_{k=1}^{K} \sum_{u \in V} \sum_{v \in N_u^{(d_k)}} |\Delta X(v)|^2 = \sum_{k=1}^{K} \sum_{u,v \in V} \mathbf{1}_{\{v \in N_u^{(d_k)}\}} |\Delta X(v)|^2$$

$$= \sum_{k=1}^{K} \sum_{u,v \in V} \mathbf{1}_{\{u \in N_v^{(d_k)}\}} |\Delta X(v)|^2 = \sum_{k=1}^{K} \sum_{v \in V} |N_v^{(d_k)}| \cdot |\Delta X(v)|^2 \le Kp \sum_{v \in V} |\Delta X(v)|^2,$$

where we used the assumption on $Kp$ to obtain the last $\le$. Then (16) continues as

$$\le (\|a\|_2 \beta^{(1)})^2 Kp \|\Delta X\|_{2,V}^2,$$

which proves that $\|\Delta Y\|_{2,V} \le (\|a\|_2 \beta^{(1)})\sqrt{Kp}\|\Delta X\|_{2,V}$ as claimed. □

*Proof of Theorem 2.* Same as in the proof of Theorem 1, we have (14). The eigen-decomposition $L_u^{(k)} = \Psi_u^{(k)} \Lambda_u^{(k)} (\Psi_u^{(k)})^T$ has that $(\Psi_u^{(k)})^T \Psi_u^{(k)} = I$, and, under the connectivity condition of the subgraph, the diagonal entries of $\Lambda_u^{(k)}$ all $> 0$. Thus

$$\langle u, v \rangle_{N_u^{(d_k)}} = \langle (\Lambda_u^{(k)})^{1/2} \Psi_u^{(k)} u, (\Lambda_u^{(k)})^{-1/2} \Psi_u^{(k)} v \rangle_{N_u^{(d_k)}},$$

which gives the Cauchy-Schwarz with weighted 2-norm as

$$|\langle B_k(\cdot, u), \Delta X(\cdot) \rangle_{N_u^{(d_k)}}| \leq \|B_k(\cdot, u)\|_{L_u^{(k)}} \cdot \|\Delta X(\cdot)\|_{(L_u^{(k)})^{-1}}. \tag{17}$$

Then similarly as in (16), using the definition of $\beta^{(2)}$ and the the condition with $\rho$, we obtain that

$$\sum_{u \in V} |\Delta Y(u)|^2 \leq (\|a\|_2 \beta^{(2)})^2 \sum_{u,k} \rho^2 \|\Delta X(\cdot)\|_{2, N_u^{(d_k)}}^2, \tag{18}$$

and the rest of the proof is the same, which gives that

$$\sum_{u \in V} |\Delta Y(u)|^2 \leq (\|a\|_2 \beta^{(2)})^2 \rho^2 K p \|\Delta X\|_{2, V}^2,$$

which proves the claim. $\qquad\square$

# B  Up/down-wind Classification Experiment

## B.1  Dataset Setup

We generate the Up/Down wind dataset on both ring graph and chain graph with 64 nodes. Every node is assigned to a probability drawn from $(0, 1)$ uniform distribution. Node with probability less than $threshold = 0.1$ will be assigned with a gaussian distribution with $std = 1.5$. Each gaussian distribution added is masked half side. Distribution masked left half is the 'Down Wind' class, distribution masked right half is the 'Up Wind' class, as shown in left plot in Fig. 3. We then sum up all half distributions from different locations in each sample. We generate 5000 training samples and 5000 testing samples.

## B.2  Model architecture and training details

**Network architectures.**

- 2-gcn-layer model:

  GraphConv(1,32)-ReLU-MaxPool1d(2)-GraphConv(32,64)-ReLU-AvgPool(32)-FC(2),

- 1-gcn-layer model:

  GraphConv(1,32)-ReLU-AvgPool(64)-FC(2),

where GraphConv can be ChebNet or L3Net.

**Training details.**

We choose the Adam Optimizer, batch size of 100, set initial learning rate of $1 \times 10^{-3}$, make it decay by 0.1 at 80 epoch and train for 100 epochs.

## B.3  Additional results

We report additional results using 1-gcn layer architecture in Tab. A.1. Our L3Net again shows stronger classification performance than ChebNet.

# C  Experimental Details

## C.1  Classification of sphere mesh data

**Spherical mesh** We conduct this experiment on icosahedral spherical mesh (Baumgardner & Frederickson, 1985). Like S2CNN (Cohen et al., 2018), we project digit image onto surface

Table A.1: results of 1-gcn layer models

| Gnn model | order | #params | ring graph Acc | chain graph Acc |
|---|---|---|---|---|
| ChebNet | L=3 | 0.2k | $50.80 \pm 0.24$ | $50.66 \pm 0.21$ |
| | L=5 | 0.3k | $51.14 \pm 0.21$ | $51.07 \pm 0.35$ |
| | L=9 | 0.4k | $51.68 \pm 0.38$ | $50.96 \pm 0.29$ |
| | L=30 | 1.1k | $51.37 \pm 0.14$ | $50.70 \pm 0.16$ |
| L3Net | 1 | 0.3k | $99.96 \pm 0.08$ | $99.67 \pm 0.12$ |
| | 0;1;2 | 0.8k | $\mathbf{99.96 \pm 0.01}$ | $\mathbf{99.92 \pm 0.01}$ |

of unit sphere, and follow Jiang et al. (2019) by moving projected digit to equator, avoiding coordinate singularity at poles.

Here, we details the subdivision scheme of the icosahedral spherical mesh we used. Start with an unit icosahedron, this sphere discretization progressively subdivide each face into four equal triangles, which makes this discretization uniform and accurate. Plus, this scheme provides a natural downsampling strategy for networks, as it denotes the path for aggregating information from higher-level neighbor nodes to lower-level center node. We adopt the following naming convention for different mesh resolution: start with level-0($L0$) mesh(i.e., unit icosahedron), each level above is associated with a subdivision. For level-$i(L_i)$, properties of spherical mesh are:

$$N_e = 30 \cdot 4 * i, N_f = 20 \cdot 4 * i, N_v = N_e - N_f + 2 \tag{19}$$

in which $N_f, N_e, N_v$ denote number of edges, faces, and vertices.

To give a direct illustration of how many nodes each level of mesh has, we list them below,

- $L0$ 12 nodes
- $L1$ 42 nodes
- $L2$ 162 nodes
- $L3$ 642 nodes
- $L4$ 2562 nodes
- $L5$ 10242 nodes

**Network architectures** We use a three-stage GNN model for this sphereMNIST, with each stage conduct convolution on spherical mesh of a specific level. Detailed architecture (suppose mesh levels used are $Li, Lj, Lk$):

Conv(1,16)$_{Li}$-BN-ReLU-DownSamp-ResBlock(16,16,64)$_{Lj}$-DownSamp-ResBlock(64,64,256)$_{Lk}$-AvgPool-FC(10),

We use the 4-stage model architecture for SphereModelNet-40, where 4 mesh levels are: $L5, L4, L3, L2$. Detailed architecture are:

Conv(6,32)$_{L5}$-BN-ReLU-DownSamp-ResBlock(32,32,128)$_{L4}$-DownSamp -ResBlock(128,128,512)$_{L3}$-DownSamp-ResBlock(512,512,2048)$_{L2}$-DownSamp-AvgPool-FC(40),

where the GraphConv(feat_in, feat_out) in above model architectures can be either Mesh Convolution layer or Graph Convolution layer, and "ResBlock" is a bottleneck module with two $1 \times 1$ convolution layers and one GraphConv layer.

**Training Details** For SphereMNIST experiments, we use batch size of 64, Adam optimizer, initial learning rate of 0.01 which decays by 0.5 every 10 epochs. We totally train model for 100 epochs.

For SphereModelNet-40 experiment, we batch size of 16, Adam optimizer, initial learning rate of 0.005 which decay by 0.7 every 25 epochs. We totally train 300 epochs.

**Results on fine mesh**

Tab. A.2 show the results of SphereMNIST and Sphere-ModelNet40 on fine meshes on the sphere. Specifically, the mesh used for SphereMNIST here is of levels $L4, L3, L2$, and the SphereModelNet-40 mesh of levels $L5, L4, L3, L2$, same as in Jiang et al. (2019).

Table A.2: Results on SphereMNIST and SphereModelNet-40 following setup in Jiang et al. (2019)

| Model | SphereMNIST Acc | SphereModelNet-40 Acc |
|---|---|---|
| S2CNN (Cohen et al., 2018) | 96.0 | 85.0 |
| UGSCNN (Jiang et al., 2019) | 99.2 | **90.50** |
| GCN | 95.8 | 87.07 |
| ChebNet($L$=4) | **99.3** | 88.05 |
| ChebNet($L$=5) | - | 88.90 |
| ChebNet($L$=6) | - | 88.70 |
| ChebNet($L$=7) | - | 88.78 |
| L3Net (1123) | 99.10 | 90.24 |
| L3Net (112) | 98.90 | 89.67 |

## C.2 FACIAL EXPRESSION RECOGNITION

**Landmarks setting** 15 landmarks are selected from the standard 68 facial landmarks defined in AAM (Cootes et al., 2001), and edges are connected according to prior information of human face, e.g., nearby landmarks on the eye are connected, see Fig. 1 (left).

**Dataset setup**

• CK+:

The CK+ dataset (Lucey et al., 2010) is the mostly used laboratory-controlled FER dataset (downloaded from: *http://www.jeffcohn.net/resources/*). It contains 327 video sequences from 118 subjects with seven basic expression labels(anger, contempt, disgust, fear, happiness, sadness, and surprise). Every sequence shows a shift from neutral face to the peak expression. Following the commonly used '(static) image-based' methods (Li & Deng, 2020), we extract the one to three frames in each expression sequence that have peak expression information in the CK+ dataset, and form a dataset with 981 image samples. Every facial image is aligned and resized to $(120, 120)$ with face alignment model (Bulat & Tzimiropoulos, 2017), and then we use this model again to get facial landmarks. As we describe in Sec. 4.2, we select 15 from 68 facial landmarks and build graph on them. The input feature for each node is an image patch centered at the landmark with size $(20, 20)$, concatenated with the landmark's coordinates, so the total input feature dimension is 402.

• FER13:

FER13 dataset (Goodfellow et al., 2013) is a large-scaled, unconstrained database collected automatically by Goole Image API (downloaded from: *https://www.kaggle.com/c/challenges-in-representation-learning-facial-expression-recognition-challenge/data*). It contains 28,709 training images, 3589 validation images and 3589 test images of size $(48, 48)$ with seven common expression labels as CK+. We align facial images, get facial landmarks, and select nodes & build graph the same way as we do in CK+. Input features are local image patch centered at each landmark with size $(8, 8)$ and landmark's coordinates, so the total input feature dimension is 66.

**Network architectures.**

• CK+:

GraphConv(402,64)-BN-ReLU-GraphConv(64,128)-BN-ReLU-FC(7),

• FER13:

GraphConv(66,64)-BN-ReLU-GraphConv(64,128)-BN-ReLU-GraphConv(128,256)-BN-ReLU-FC(7),

where GraphConv(feat_in, feat_out) here can be any type of graph convolution layer, including our L3Net.

**Training details.**

• CK+:

We use 10-fold cross validation as Ding et al. (2017). Batch size is set as 16, learning rate is 0.001 which decay by 0.1 if validation loss remains same for last 15 epochs. We choose Adam optimizer and train 100 epochs for each fold validation.

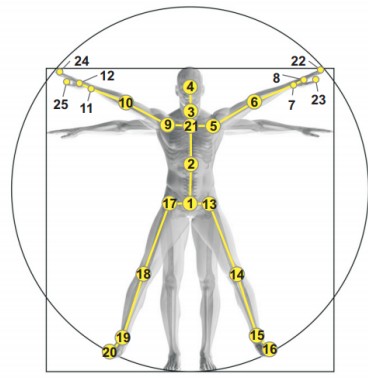

Figure A.2: Illustration of 25-point body joints and graph.

- FER13:

We report results on test set. Batch size is set as 32, learning rate is 0.0001 which decay 0.1 if validation loss remains same for last 20 epochs. We choose Adam optimizer and train models for 150 epochs.

**Runtime analysis details.** In section 4.2, we report the running time of our L3Net(order 1,1,2,3), 13.02ms, and best ChebNet, 12.56ms, on CK+ dataset, which are comparable. Here, we provide more details about this. The time we use to compare is the time of model finishing inference on validation set with batch size of 16. For each model, we record all validation time usages in all folds and report the average of them. The Runtime analysis is performed on a single NVIDIA TITAN V GPU.

### C.3    Skeleton-based Action Recognition

**Dataset setup.**

- NTU-RGB+D:

NTU-RGB+D (Shahroudy et al., 2016) is a large skeleton-based action recognition dataset with three-dimensional coordinates given to every body joint (downloaded from: *http://rose1.ntu.edu.sg/datasets/requesterAdd.asp?DS=3*). It comprises 60 action classes and total 56,000 action clips. Every clip is captured by three fixed Kineticsv2 sensors in lab environment performed by one of 40 different subjects. Three sensors are set at same height but in different horizontal views, $-45°, 0°, 45°$. There are 25 joints tracked, as shown in Fig. A.2. Two experiment setting are proposed by Shahroudy et al. (2016), cross-view (X-view) and cross-subject (X-sub). X-view consists of 37,920 clips for training and 18960 for testing, where training clips are from sensor on $0°, 45°$, testing clips from sensor on $-45°$. X-sub has 40,320 clips for training and 16,560 clips for testing, where training clips are from 20 subjects, testing clips are from the other 20 subjects. We test our model on both settings.

- Kinetics:

Kinetics (Kay et al., 2017) is a large and most commonly-used action recognition dataset with nearly 300,000 clips for 400 classes (downloaded from: *https://deepmind.com/research/open-source/kinetics*). We follow Yan et al. (2018) to get 18-point body joints from each frame using OpenPose (Cao et al., 2017) toolkit. Input features for each joint to the Network is $(x, y, p)$, in which $x, y$ are 2D coordinates of the joint, and $p$ is the confidence for localizing the joint. To eliminate the effect of skeleton-based model's inability to recognize objects in clips, we mainly focus on action classes that requires only body movements. Thus, we conduct our experiments on Kinetics-Motion, proposed by Yan et al. (2018). This is a small dataset that contains 30 action classes strongly related to body motion. Note that there are severe data missing problem in landmark coordinates in Kinetics data, so we also use our regularization scheme in this experiment.

**Network Architectures.**

- NTU-RGB+D:

We follow the architecture in Yan et al. (2018):

STGraphConv(3,64,9,s1)-STGraphConv(64,64,9,s1)-STGraphConv(64,64,9,s1)-
STGraphConv(64,64,9,s1)-STGraphConv(64,128,9,s2)-STGraphConv(128,128,9,s1)-
STGraphConv(128,128,9,s1)-STGraphConv(128,256,9,s2)-STGraphConv(256,256,9,s1)-
STGraphConv(256,256,9,s1)-STAvgPool-fc(60).

- Kinetics:

We also design a computation-efficient architecture for Kinetics-Motion with larger temporal downsampling rate, which results in less forward time:

STGraphConv(3,32,9,s2)-STGraphConv(32,64,9,s2)-STGraphConv(64,64,9,s1)-
STGraphConv(64,64,9,s1)-STGraphConv(64,128,9,s2)-STGraphConv(128,128,5,s1)-
STGraphConv(128,128,5,s1)-STGraphConv(128,256,5,s2)-STGraphConv(256,256,3,s1)-
STGraphConv(256,256,3,s1)-STAvgPool-fc(60),

where the structure of STGraphConv(feat_in, feat_out, temporal_kernel_size, temporal_stride) is:

GraphConv(feat_in, feat_out)-BN-ReLU-1DTemporalConv(feat_out, feat_out, temporal_kernel_size, temporal_stride)-BN-ReLU.

**Training Details**

- NTU-RGB+D:

We use batch size of 32, initial learning rate of 0.001 which decay by 0.1 at (30, 80) epoch, and total train 120 epochs. SGD optimizer is selected. We padding every sample temporally with 0 to 300 frames.

- Kinetics:

We use batch size of 32, initial learning rate of 0.01 which decay by 0.1 at (40, 80) epoch, and total train 100 epochs. SGD optimizer is selected. We padding every sample temporally with 0 to 300 frames, and during training, we perform data augmentation by randomly choosing 150 contiguous frames.

C.4 DETAILS OF EXPERIMENT ON MNIST

C.4.1 SIMULATED GRAPH NOISE ON $7 \times 7$ MNIST.

Here we describe three types of noise in our experiments:

**Gaussian noise**. Given a $7 \times 7$ image from MNIST, we sample 49 values from $\mathcal{N}(0, std^2)$. The $std$ controls the strength of noise added. We conduct experiments under $std = 0.1, 0.2, 0.3$ as shown in Tab. 3. The amount of noise is also measured by PNSR which is standard for image data.

**Missing value noise.** Given a image, we randomly sample 49 values from $U(0, 1)$, and select nodes with probabilities less than a threshold. This threshold is called $noise\_level$, which controls the percentage of nodes affected. Then, we remove the pixel value at those selected nodes. Experiments with $noise\_level = 0.1, 0.2, 0.3$ are conducted.

**Graph node permutation noise**. For each sample, we randomly select a permutation center node which has exact 4 neighbors. Then, we rotate its neighbors clockwise by 90 degree, e.g., top neighbor becomes right neighbor, and then we update the indices of permuted nodes.

Table A.3: Results on MNIST with grid size $28 \times 28$, L3Net-pooling uses graph pooling between convolutional layers.

| Model | bases order | #params (w/o FC) | Acc |
|---|---|---|---|
| GCN | 1 | 2.4k | $93.30 \pm 0.12$ |
| ChebNet | $L=3$ | 6.5k | $93.93 \pm 0.18$ |
| | $L=4$ | 8.6k | $94.97 \pm 0.06$ |
| | $L=5$ | 10.7k | $95.87 \pm 0.09$ |
| | $L=6$ | 12.8k | $96.64 \pm 0.12$ |
| | $L=7$ | 14.8k | $96.98 \pm 0.19$ |
| | $L=9$ | 19.0k | $97.43 \pm 0.14$ |
| | $L=15$ | 31.5k | $\mathbf{97.91 \pm 0.08}$ |
| | $L=20$ | 41.9k | $97.90 \pm 0.04$ |
| L3Net | 1;1;2 | 41.0k | $96.78 \pm 0.08$ |
| | 1;1;2;3 | 79.2k | $97.32 \pm 0.10$ |
| L3Net-pooling | 1;1;2 | 27.9k | $97.11 \pm 0.09$ |
| | 1;1;2;3 | 52.2k | $97.54 \pm 0.07$ |

Table A.4: Results on MNIST with grid size $14 \times 14$

| Model | bases order | #params (w/o FC) | Acc |
|---|---|---|---|
| GCN | 1 | 2.4k | $93.70 \pm 0.09$ |
| ChebNet | $L=3$ | 6.5k | $96.06 \pm 0.16$ |
| | $L=4$ | 8.6k | $96.85 \pm 0.11$ |
| | $L=5$ | 10.7k | $97.24 \pm 0.28$ |
| | $L=6$ | 12.8k | $97.58 \pm 0.10$ |
| | $L=7$ | 14.9k | $\mathbf{97.74 \pm 0.07}$ |
| L3Net | 0;1;2 | 13.3k | $97.17 \pm 0.09$ |
| | 1;1;2 | 14.8k | $97.24 \pm 0.12$ |
| | 1;1;2 reg0.001 | 14.8k | $97.43 \pm 0.07$ |
| | 1;1;2;3 | 25.1k | $97.51 \pm 0.07$ |

Table A.5: Results on MNIST with grid size $7 \times 7$ with different levels of missing value

| Model | bases order | reg | #params (w/o FC) | Acc(original) | Acc(psnr 18.70) | Acc(psnr 15.33) | Acc(psnr 13.15) |
|---|---|---|---|---|---|---|---|
| GCN | 1 | - | 2.4k | $90.02 \pm 0.24$ | $83.44 \pm 0.15$ | $77.23 \pm 0.13$ | $71.67 \pm 0.06$ |
| ChebNet | $L=3$ | - | 6.5k | $92.85 \pm 0.09$ | $87.09 \pm 0.18$ | $82.11 \pm 0.18$ | $76.15 \pm 0.26$ |
| | $L=4$ | - | 8.6k | $93.12 \pm 0.1$ | $87.09 \pm 0.16$ | $82.22 \pm 0.28$ | $75.95 \pm 0.22$ |
| | $L=5$ | - | 10.7k | $93.2 \pm 0.07$ | $87.01 \pm 0.14$ | $82.04 \pm 0.14$ | $76.21 \pm 0.38$ |
| | $L=6$ | - | 12.7k | $93.42 \pm 0.09$ | $87.20 \pm 0.3$ | $81.19 \pm 0.29$ | $75.24 \pm 0.32$ |
| | $L=7$ | - | 14.8k | $93.45 \pm 0.06$ | $87.08 \pm 0.11$ | $81.00 \pm 0.17$ | $75.31 \pm 0.34$ |
| **L3Net** | 1;1;2 | - | 8.4k | $93.56 \pm 0.08$ | $86.64 \pm 0.16$ | $81.14 \pm 0.30$ | $75.07 \pm 0.08$ |
| | 1;1;2 | 0.5 | 8.4k | $93.85 \pm 0.13$ | $87.22 \pm 0.23$ | $\mathbf{82.84 \pm 0.11}$ | $\mathbf{76.48 \pm 0.23}$ |
| | 1;1;2;3 | - | 12.2k | $93.67 \pm 0.15$ | $86.51 \pm 0.38$ | $80.68 \pm 0.11$ | $74.24 \pm 0.36$ |
| | 1;1;2;3 | 0.5 | 12.2k | $\mathbf{93.85 \pm 0.15}$ | $\mathbf{87.22 \pm 0.08}$ | $82.64 \pm 0.31$ | $76.08 \pm 0.38$ |

## C.4.2 NETWORK ARCHITECTURE AND TRAINING DETAILS

We use the same architecture for different experiment settings:

GraphConv(1,32)-BN-ReLU-GraphConv(32,64)-BN-ReLU-FC(10),

where GraphConv can be different types of graph convolution layers. We set batch size to 100, use Adam optimizer, and set initial learning rate to 1e-3. Learning rate will drop by 10 if the least validation loss remains the same for the last 15 epochs. We set total training epochs as 200. We use 10,000 images for training.

We also adopt graph pooling layers in the above architecture:

GraphConv(1,32)-BN-ReLU-Graph Pooling-GraphConv(32, 64)-BN-ReLU-
Graph Pooling-FC(10).

More discussion about graph pooling layer and multi-scale graph convolution is detailed in Appendix C.5.

## C.4.3 ADDITIONAL RESULTS

Here, we show experiments results on $28 \times 28, 14 \times 14$ grid, as well as $7 \times 7$ grid with missing values. Tab. A.3 shows results on $28 \times 28$ image grid. Our model have better performance than other methods.

Tab. A.4 shows results on $14 \times 14$ image grid, where our L3Net have comparable results with the best ChebNet (Defferrard et al., 2016) method.

We shows our results on $7 \times 7$ image grid with missing values in Tab. A.5. With regularization, L3Net achieves the best performance in every experiment with different noise levels.

## C.5 MULTI-SCALE GRAPH CONVOLUTION

The proposed L3Net graph convolution model (2) is compatible with graph down/up-sampling schemes to achieve multi-scale feature extraction.

The graph down/up-sampling is usually implemented as a separate layer between graph convolution layers. As an example, Fig. A.3 illustrates three levels of graphs produced from an originally $14 \times 14$ image grid, denoted as $G_1$, $G_2$, $G_3$, and they have 176, 45 and 9 nodes

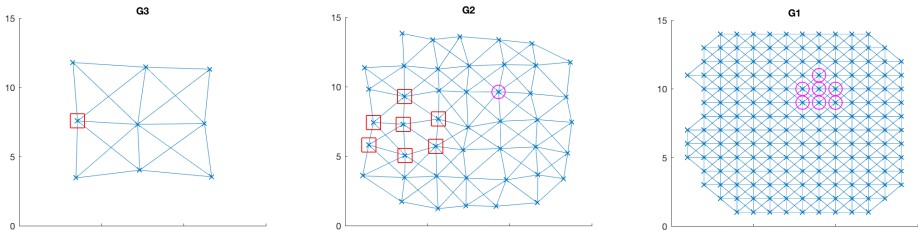

Figure A.3: Three levels of graphs on a 14×14 image grid. From left to right: the top level $G_3$ (the coarsest) to the bottom level $G_1$ (the finest). The red square (pink circle) indicates the node $x$ on $G_3$ ($G_2$) and its local neighborhood on $G_2$ ($G_1$), on which the graph pooling is applied.

respectively. On $G_1$, 10% of pixels which contain the lowest amount of pixel intensities over the dataset are removed, and those nodes are located near the boundary of the canvas. For each node $x_i'$ in the coarse-grained graph $G_2$, a neighborhood consisting of nodes in $G_1$ is constructed, called $N(x_i'; G_1)$. A pooling operator computes the feature on $x_i'$ from those on $N(x_i'; G_1)$, and the pooled feature is used as the input to the graph convolution on $G_2$. A similar graph pooling layer is used from $G_2$ to $G_3$. The graph topology and local neighborhoods are determined by grid point locations. Using a two-layer convolution with graph poolings in between from $G_1$ to $G_3$, and the other setting same as in Table A.4, L3Net obtains $97.33 \pm 0.15$ test accuracy (basis order $1; 1; 2$, with regularization 0.001).

We have also applied graph pooling layers on regular image grid on the 28×28 MNIST dataset. The results, reported in Table A.3, show that multi-scale convolution in L3Net not only improves the classification accuracy but also reduces the number of parameters.

Graph up-sampling layer can be used similarly. These multi-scale approaches generally apply to graph convolution models, see, e.g., the hierarchical construction originally proposed for locally-connected GNN (Coates & Ng, 2011; Bruna et al., 2013). There is also flexibility in defining the graph down/up-sampling schemes, and the choice depends on application. An example of graph sampling operator on face mesh data is given in (Ranjan et al., 2018). At last, apart from using separate down/up-sampling layers, it is also possible to extend the L3Net model (2) to directly implement graph down/up-sampling, which would be similar to the convolution-with-stride (conv-t) operator in standard CNN. Specifically, between $G_l$ and $G_{l+1}$, the local basis filter $B_k(u, u')$ is defined for $u' \in G_l$ and $u \in G_{l+1}$, and $B_k(u, u') \neq 0$ only when $u'$ is in a local neighborhood of $u$. In matrix notation, $B_k$ is of size $|G_{l+1}|$-by-$|G_l|$, and is sparse according to the graph local neighborhood relation between $G_l$ and $G_{l+1}$.

