# OpenReview forum: "Graph Convolution with Low-rank Learnable Local Filters"
_ICLR.cc/2021/Conference — ICLR 2021 Spotlight_

### Official Review · AnonReviewer4 · 2020-10-27
**Good general model for graph convolution**

**Rating:** 7
**Confidence:** 3

**Review:**

The authors present a new definition of graph convolution that is also shown to generalize well-known existing ones.

Pros:
- interesting new definition of graph convolution
- good theoretical contribution

Cons:
- experiments could have used more widely adopted benchmarks

The novel contribution of the paper is sound and the theoretical explanation allows to understand the connections with existing graph convolution definitions.

The experiments are well conducted and I appreciate the indication of the standard deviation in the results. They also show significant gains with respect to the other techniques. While the currently reported experiments are adequate, it would have been more interesting to test the method on emerging benchmarking frameworks such as [1] to get better insights on the perfromance on a standardized setting.

[1] Benchmarking Graph Neural Networks,
Vijay Prakash Dwivedi, Chaitanya K. Joshi, Thomas Laurent, Yoshua Bengio, Xavier Bresson, https://arxiv.org/abs/2003.00982

---

### Official Review · AnonReviewer3 · 2020-10-28
**This paper proposes a graph neural network method that uses a convolution operation with decomposed, low-rank filters. Robustness against data perturbation or missing data is fulfilled by local graph regularization.**

**Rating:** 7
**Confidence:** 3

**Review:**

####################
Pros:
$\bullet$ The proposed graph convolution method is tested on different problems (object recognition on the spherical mesh, facial expression recognition, action recognition on the face and body landmarks), and real-world datasets.

$\bullet$ The proposed method performs well both under the missing node and feature and graph noise. Particularly the effect of local graph Laplacian regularization is notable.

$\bullet$ The relationship between the proposed L3Net and previous spatial and spectral graph convolutions is theoretically explained.

$\bullet$ The complexity of the L3Net is significantly lower than locally connected GNN, and it is more suitable for small-size graphs such as the face or body keypoints.

####################
Cons:
$\bullet$ In face experiments, pixel values around each node were taken as node features, however, in a similar setting in action recognition, they were not used. Considering the temporal data as in CK+, applying a similar setting by changing ST-GCN blocks with L3Net would make it comparable.

####################
Minor:
$-$ In many parts, typo error in MNIST dataset's name ("MNSIT") .
$-$ 2.3 first paragraph, graph "convolutoins"-> convolutions.
$-$ 1.1. second paragraph: "Chebyshev polynomial in Chenbet"-> "ChebNet or ChebConv"

---

### Official Review · AnonReviewer2 · 2020-10-29
**This paper proposed L3Net which is a new graph convolution with low-rank learnable local filters. L3Net is applicable for both spatial and spectral graph convolution. Experiments are conducted on mesh data, facial recognition and action recognition. Its robustness to graph noise is also tested.**

**Rating:** 7
**Confidence:** 3

**Review:**

This paper proposed L3Net which is a new graph convolution with decomposing the learnable local filters into low-rank. It can contain both spatial and spectral graph convolution (including ChebNet, GAT, EdgeNet and so on) as subsets. It is also robust to graph noise. Experiments are conducted on mesh data, facial recognition and action recognition, indicating out-performed performance over baselines. Its robustness to graph noise is also tested.

In general, the motivation, novelty and validation are good. However, I have the following concerns:

Although the authors demonstrate and explain the advantage and disadvantages of L3Net, the authors do not explain the applicability of down-sampling and up-sampling in L3Net. For example, following max-pooling and up-sampling in DCNN, ChebNet can be integrated into mufti-scales for enhancing the performance (see https://arxiv.org/pdf/1807.10267.pdf ), how about formulating L3Net into multi-scales?
In the experiments, I saw that three convolutional layers are used in Section 4.1 (4.2, 4.3 and 4.4 seems do not offer the depth information of L3Net). I feel that three is a really shallow network. Why deeper networks are not used? Can the authors kindly comment/validate on this please?
In Proposition 1, it seems that a control of “K” and “L” can offer different options between ChebNet, GAT and EdgeNet. In practical training, are these settings of “K” and “L” easy to train/converge, with always good performance and computationally light? Are validation offered to prove this proposition and to compare the different options offered by L3Net to ChebNet, GAT and EdgeNet? If not, can the authors give reasonable comments on this please?
It seems the regularization in Section 2.2 is a very important contribution of this paper, however, no ablation study is offered (comparison between with or without regularization). Can the authors kindly validate this please?
In Figure 1, the definition of variable \mu, c, k, M and K are not defined which makes the figure less easy to understand. Could the authors kindly add these on please?
In Figure 1, the authors use “left” and “right” to distinguish the four figures, this is kind of confusing. Could the authors use a, b, c and d for the figure label please? The same for Figure 2 where the authors use “plots” and “table” to distinguish different subplots. I think this can also be improved in similar way. Also other figures.
In Section 2.1, |V|=n. I feel that “N” would be more appropriate here, as it represents the number of nodes.

---

### Official Review · AnonReviewer1 · 2020-10-30
**GNN architecture with learnable low-rank filters: an attempt to unify recent architectures and have some theoretical results**

**Rating:** 8
**Confidence:** 5

**Review:**

SUMMARY:

The paper presents a graph neural network (GNN) architecture with learnable low-rank filters that unifies various recently-proposed GNN-based methods. The local filters substitute the graph shift operator (GSO) by a learnable set of parameters that capture the local connectivity of each node in the graph. Moreover, a regularization penalty is proposed to increase the robustness of the model and prevent these local structures to overfit. The paper provides proofs to justify the generality of the approach and how different methods can be seen as a particularization of the proposed scheme. Two theorems are also proved to claim the stability of the GNN architecture against dilation perturbations in the input signal. Several numerical experiments are conducted to empirically test the usefulness of the model.


STRONG POINTS:

The paper introduces a new GNN-based approach with larger discriminability power.

The proposed approach generalizes various previously existent architectures. This is proved in the appendices.

A regularization technique is proposed to avoid overfitting to local particularities of the data.

Two theorems are introduced to proof the stability of the network against dilation in the input signal.

The numerical experiments are extensive and convincing. This is one of the strongest points of the paper.

The paper is well-structured and the style is appropriate. It is easy-to-follow and the points are clearly stated.


WEAK POINTS:

Replacing the GSO with a set of learnable parameters increases the discriminability power of the network, at the cost of sacrificing various properties of common GNN architectures. For example, this technique is no longer permutation equivariant, and transferability to larger networks will no longer be an option, as long as the learnt parameters are graph-dependant.

Scalability problems appear when the network grows in size. This, and some possible ways of tackling it, are discussed in the conclusion.

Although the theorems offer insights on the robustness of the network against perturbations on the input signal, they are restricted to dilation perturbations (which are proportional to the input signal). This is not commonly the case, perturbations often follow distributions that has nothing to do with the input signals.


OVERALL ASSESMENT AND RECOMMENDATION:

The paper introduces a new architecture with larger discriminative power that generalizes various state-of-the-art methods. Although the theoretical results are not particularly strong, they are undoubtedly insightful. Then, the empirical performance of this technique is exhaustively validated through several experiments. Thus, in my opinion, this paper should be accepted.


RECOMMENDATIONS TO THE AUTHORS:

Using matrix notation would be helpful to clarify various equations and capture the attention of a broader range of researchers. Equally important, it will also contribute to establish links between the schemes proposed in the paper and well-established techniques in the field of graph signal processing (GSP).

Describing the connection between the operator $B_k$ and classical graph operators (adjacency, Laplacian, normalized Laplacian...) would be clarifying. Consider adding a couple of lines pointing out this relation.

---

### Decision · Program_Chairs · 2021-01-07
**Final Decision**

**Decision:**

Accept (Spotlight)

**Comment:**

All reviewers expressed consistent enthusiasm on this submission during the review process. No reviewers expressed concerns and objections to accept this submission during discussion. It is quite clear this is a strong submission and deserves accept.